# p43, a Truncated Form of Thyroid Hormone Receptor α, Regulates Maturation of Pancreatic β Cells

**DOI:** 10.3390/ijms22052489

**Published:** 2021-03-02

**Authors:** Emilie Blanchet, Laurence Pessemesse, Christine Feillet-Coudray, Charles Coudray, Chantal Cabello, Christelle Bertrand-Gaday, François Casas

**Affiliations:** DMEM (Dynamique du Muscle et Métabolisme), INRAE, University Montpellier, 34060 Montpellier, France; blanchet.emilie@yahoo.fr (E.B.); laurence.pessemesse@inrae.fr (L.P.); christine.coudray@inrae.fr (C.F.-C.); charles.coudray@inrae.fr (C.C.); chantal.cabello@inrae.fr (C.C.); christelle.bertrand-gaday@inrae.fr (C.B.-G.)

**Keywords:** thyroid hormone, p43, mitochondria, beta-cells, insulin, MafA, diabetes

## Abstract

P43 is a truncated form of thyroid hormone receptor α localized in mitochondria, which stimulates mitochondrial respiratory chain activity. Previously, we showed that deletion of p43 led to reduction of pancreatic islet density and a loss of glucose-stimulated insulin secretion in adult mice. The present study was designed to determine whether p43 was involved in the processes of β cell development and maturation. We used neonatal, juvenile, and adult p43-/- mice, and we analyzed the development of β cells in the pancreas. Here, we show that p43 deletion affected only slightly β cell proliferation during the postnatal period. However, we found a dramatic fall in p43-/- mice of MafA expression (V-Maf Avian Musculoaponeurotic Fibrosarcoma Oncogene Homolog A), a key transcription factor of beta-cell maturation. Analysis of the expression of antioxidant enzymes in pancreatic islet and 4-hydroxynonenal (4-HNE) (a specific marker of lipid peroxidation) staining revealed that oxidative stress occurred in mice lacking p43. Lastly, administration of antioxidants cocktail to p43-/- pregnant mice restored a normal islet density but failed to ensure an insulin secretion in response to glucose. Our findings demonstrated that p43 drives the maturation of β cells via its induction of transcription factor MafA during the critical postnatal window.

## 1. Introduction

Thyroid hormone is a major regulator of metabolism and mitochondrial function [1,2] and also a key regulator of the postnatal maturation of many tissues. Thyroid hormone also affects different metabolic aspects of glucose and insulin metabolism. Hypothyroidism is associated with a decrease of normal glucose-stimulated insulin secretion by the beta-cells [3], while an increase of insulin secretion has been reported in hyperthyroidism [4]. In addition, Ligget and coworkers have observed an increase in insulin secretory rate after T3 administration in rats [5]. Thyroid hormone acts through nuclear receptors (T3Rs) encoded by the TRα and TRβ genes [6,7]. TRα and TRβ have been identified in adult islet cells [8,9]. Interestingly, a comparison of the expression of the isoforms shows that TRα predominates at early ages in β cells, whereas TRβ becomes the predominant isoform in adult islets [8]. Recently, Aguayo-Mazzucato and coworkers [10] have shown that thyroid hormone induces both maturation and aging effectors in β cells. Through direct binding on the promoter of the genes, they showed that TRα enhances *p16*^Ink4a^ expression, a β cells senescent marker and effector, whereas TRβ drive the expression of MafA, a key transcription factor driving the maturation of the insulin secretory response to glucose in neonatal β cells. Moreover, Furuya et al. [11] have shown that liganded TRα enhances the proliferation of pancreatic β cells. The observation that thyroid hormone also regulates pancreatic islet maturation during zebrafish development suggests that the role of thyroid hormone in the functional maturation of β cells is preserved across species [12].

Previously, we have identified a truncated form of the nuclear receptor TRα1, with a molecular mass of 43 kDa (p43), which stimulates mitochondrial transcription and protein synthesis in the presence of T3 [13,14]. This protein is synthesized by the use of the internal initiation site of translation occurring in the TRα1 transcript and localized in mitochondria. This receptor, which stimulates mitochondrial respiratory chain activity, is expressed ubiquitously, but more particularly in the most metabolically active tissues [13,14]. It is notably involved in the regulation of skeletal muscle phenotype in adulthood [15,16,17] and during regeneration [18]. We also demonstrated recently that its absence induced an exacerbated age-related hearing loss in mice [19]. To assess the physiological importance of p43, we generated mice carrying specific p43 invalidation but that still express other TRα proteins [20]. We reported that p43 depletion in mice induced a decrease of the mitochondrial respiratory chain activities in the pancreatic islet, a reduction of islet density and a loss of glucose-stimulated insulin secretion [20]. In addition, during aging, p43 depletion in mice progressively induced all the characteristics of type 2 diabetes (hyperglycemia, glucose intolerance and insulin resistance) [21].

In the present study, our aim was to determine whether p43 was involved in the processes of β cell development and maturation. Because the postnatal period is a critical window for the maturation and the acquisition of glucose-stimulated insulin secretion [22,23,24], we used neonatal, juvenile and adult p43-/- mice, and we analyzed the development of the β cells in the pancreas.

## 2. Results

### 2.1. β Cells Proliferation is Slightly Affected in Mice Lacking p43

As previously reported [20,21], histological analysis revealed a strong decrease of islet density in 3-month old mice lacking p43 (−40%, *p* < 0.05) (Figure 1A). However, the islet density in p43-/- neonates was similar to the controls (Figure 1A). To investigate the mechanism responsible for the decreased growth of the p43-/- islets, we compared the proliferation index of β cells in the pancreas using antibodies raised against insulin and PH3 (Phosphohistone H3), a mitotic marker. In the pancreas from WT mice, we observed a progressive decrease of proliferation with age, whereas, in the pancreas of p43-/- mice, the peak of proliferation was reached at 7 days before decreasing. Compared to WT, in p43-/- pancreas, we found a decreased proliferation of insulin+ progenitor cells at 2 days (−72%, *p* < 0.001) and an increase at 7 days ( + 57%, *p* < 0.05), and no difference beyond that (Figure 1B,C). Together, these data demonstrate that p43-/- deletion slightly affects the proliferation of β cells in neonatal mice.

### 2.2. Deletion of p43 Alters Insulin Content and the α/β Cell Ratio during the Postnatal Period

We next evaluated the influence of p43 deletion on insulin content by measuring the immunofluorescent labeling intensity using an antibody raised against insulin. Our quantitative results showed a strong decrease of insulin content in p43-/- β cells in comparison to wild-type pancreas at 2, 7 and 14 days (−26%, *p* < 0.001 at 2 days; −65%, *p* < 0.001 at 7 days; −59%, *p* < 0.001 at 14 days), whereas the content appeared normal in adult pancreas (Figure 2A,B). However, after Q-PCR performed on islets isolated by laser microdissection, we did not observe any difference in the expression of Ins1 and Ins2 (the 2 genes coding for insulin [25]) in the absence of p43 (Figure 2C,D). 

To address the ratio of α cells to β cells (α/β ratio), we performed immunofluorescent labeling using antibodies raised against insulin and glucagon. We found that the α/β ratio was higher in p43-/- pancreas at 2 and 7 days (+49%, *p* < 0.01 at 2 days; +67%, *p* < 0.05 at 7 days) whereas the proportion of alpha cells and beta-cells remained similar to the controls in older mice (Figure 2C,D). Together, these data demonstrate that p43-/- deletion affects insulin content and α/β ratio during the postnatal period, but not in adult mice.

### 2.3. Deletion of p43 Strongly Decreases MafA Expression

To better understand the loss of glucose-stimulated insulin secretion previously recorded in p43-/- mice, we isolated the islets by laser microdissection, and then we analyzed by Q-PCR various key transcription factors involved in pancreas development and β cells differentiation and maturation. MafA [26] regulates β cells maturation and the acquisition of glucose-stimulated insulin secretion in vivo. PDX1 (pancreas/duodenum homeobox protein 1), also known as insulin promoter factor 1, is a transcription factor that plays a crucial role in pancreatic development and beta-cell maturation [27]. MafB (V-maf musculoaponeurotic fibrosarcoma oncogene homolog B) is a transcription factor critical to β cell terminal differentiation, but it progressively disappears from β cells after birth and becomes a specific factor of adult α cells [28]. Aristaless related homeobox (ARX), NeuroG3 (neurogenin 3), neurogenic differentiation 1 (NeuroD1) and Nkx6.1 are transcription factors that play a key role in pancreatic endocrine fate specification [27,29,30]. We found that MafA mRNA expression remained dramatically low in p43-/- islet whereas it gradually increased in controls (−49%, *p* < 0.05 at 2 days; −56%, *p* < 0.05 at 7 days; −58%, *p* < 0.01 at 14 days; −60%, *p* < 0.01 at 3 months) (Figure 3A). In contrast, the mRNA expression of PDX1, MafB, ARX, neurogenin 3, NeuroD1 and Nkx6.1 was not altered by p43 deletion (Figure 3B,G).

We further explored the expression of MafA in immunofluorescent experiments. Surprisingly, we failed to detect MafA expression at 2 and 7 days in both genotypes. At 14 days, we found that MafA was expressed in only 32% of beta-cells in p43-/- pancreas against 42% in controls (*p* < 0.001) (Figure 4A,B). At 3 months, MafA was expressed in 67% of β cells in controls and in only 60% in p43-/- (*p* < 0.01) (Figure 4A,B). Next, we evaluated the MafA level by measuring the immunohistochemical labeling intensity. Our quantitative results revealed a strong decrease of MafA expression at 14 days and at 3 months in p43-/- pancreas (−35%, *p* < 0.001 at 14 days; −42%, *p* < 0.001 at 3 months) (Figure 4C). These results showed that in p43-/-, the MafA protein is expressed in fewer beta-cells and in smaller amounts.

Although PDX1 was not affected at mRNA level by p43 deletion, we investigated its expression in immunofluorescent experiments because it is a major regulator of insulin gene expression, and it plays a dominant role in maintaining beta-cell identity [31]. We found that in both genotypes, the percentage of PDX1 expression in β cells progressively increased with age from 42% at 2 days to around 80% in adult mice (Figure 4D,E). Next, we evaluated the PDX1 level by measuring the immunohistochemical labeling intensity. Our quantitative results revealed a strong increase of PDX1 expression in the postnatal period in p43-/- pancreas (+99%, *p* < 0.001 at 2 days; +51%, *p* < 0.001 at 7 days; +91%, *p* < 0.001 at 14 days) whereas its level slightly decreased in adult (−19%, *p* < 0.001 at 3 months) (Figure 4F).

### 2.4. Deletion of p43 Induces a Strong Oxidative Stress

We have previously shown that a defect of p43 expression in skeletal muscle increases ROS production leading to oxidative stress [15,32]. To examine this possibility at the beta-cell level, we performed immunohistochemical experiments using an antibody raised against 4-hydroxynonenal (4-HNE), a specific marker of lipid peroxidation. At 2 days, we observed a strong accumulation of 4-HNE in p43-/- islet in comparison to WT, whereas no difference between the genotypes was seen in the older pancreas (Figure 5A). This observation prompted us to investigate the expression of four antioxidant enzymes by Q-PCR in the islets. At 2 days, we found no differences in the mRNA expression of glutathione peroxidase (GPX), catalase, superoxide dismutase 1 and 2 (SOD1 and SOD2) (Figure 5B,E). However, from 7 days, a strong increase of GPX and SOD2 was recorded in p43-/- islet, whereas expression of catalase and SOD1 remained unchanged (Figure 5B,E). Together, these results suggest that the oxidative stress detected in the p43-/- islets at 2 days is subsequently offset by an increase in the expression of the antioxidant enzymes GPX and SOD2.

### 2.5. Antioxidant Treatment Partially Reverses the Pancreatic Phenotype of p43-/- Mice

To further analyze the implication of ROS in the reduction of islet density and in a loss of glucose-stimulated insulin secretion, p43-/- pregnant mice received a combination of antioxidants until weaning.

At 3 months of age, insulin levels of offspring were measured in plasma after overnight food withdrawal or 30 min after an oral glucose overload. As expected, the glucose overload induced a significant increase of insulinemia in WT mice, whereas we found that insulin levels remained unchanged after glucose overload in p43-/- mice whether or not they received an antioxidants supplementation (Figure 6A). In contrast, histological analysis revealed that the antioxidant supplementation abolished the decrease of islet density previously observed in 3-months old mice lacking p43 (Figure 6B). These data demonstrated that antioxidants supplementation in p43-/- mice partially restores a normal pancreatic β cells phenotype.

## 3. Discussion

We had previously shown that p43 depletion in mice led to a reduction of islet density and a loss of glucose-stimulated insulin secretion [20,21]. Here we have examined the relationship between the absence of p43 and islet β cells formation and function during the postnatal period, a critical window for the maturation and the acquisition of glucose-stimulated insulin secretion [22,23,24].

We showed that the islet density in p43-/- neonates were similar to the controls. In addition, the deletion of p43 affects only slightly proliferation of β cells during the postnatal period, which cannot explain the decrease in islet density observed in adult mice. These data indicate that the mechanisms involved in the reduction of islet density in adulthood in p43-/- mice occur after the neonatal period studied here. Moreover, the finding that antioxidant supplementation during pregnancy until weaning restores a normal islet density in offspring p43-/- mice suggests that the proliferation defect is at least in part mediated by the activation of ROS production.

We hypothesized that the lack of insulin secretion in response to glucose recorded in the absence of p43 results from a generalized low expression of genes characteristic of mature functional β cells. Among these genes, we focused our attention on two glucose-responsive transcription factors, MafA and Pdx1, known to regulate genes involved in insulin synthesis and secretion. In particular, studies conducted over the past decade by several groups strongly support that MafA [26] regulates β cells maturation and the acquisition of glucose-stimulated insulin secretion in vivo [33,34,35,36,37]. Interestingly, we found that MafA expression both at mRNA and protein levels remained dramatically low in p43-/- β cells in comparison to WT mice. Because MafA is known to induce endogenous insulin transcription [36], it was not a surprise to observe a decrease of insulin content, at least during the postnatal period in mice lacking p43. However, while the MafA messenger is well expressed at 2 and 7 days in both genotypes, we were unable to detect the protein by immunofluorescence at these two stages even in C57BL6/J mice in contrast to a previous publication [34]. Moreover, despite the fact that we used the same antibody, we observed that MafA is expressed in only 42% and 67% of β cells at 14 days and 3 months in WT mice, whereas Artner and coworkers found an expression level of 79% and 81% at the same stage [34]. The differences we observed are probably due to a difference in the breeding conditions (temperature, diet and enrichment).

Our results also revealed an increase of PDX1 expression at the protein level in the p43-/- pancreas during the neonatal period. Because overexpression of PDX1 is also known to increase insulin content [33,38], we postulate that this increase of PDX1 is most likely an adaptation to compensate in part for the decrease in insulin synthesis in beta-cells in response to the fall of MafA expression. However, as previously shown, this overexpression of PDX1 in neonatal islets was unable to stimulate insulin secretion in response to glucose [33]. This set of data demonstrates that p43 is a physiological regulator of functional maturation of beta-cells via its induction of MafA.

Mitochondria have emerged as central players in the regulation of adult beta-cell function [39,40]. We have previously shown that p43 deletion in mice induced a decrease of the mitochondrial respiratory chain activities and abolished beta-cell maturation [20]. Yoshihara and coworkers [41] have shown that an orphan nuclear receptor, Estrogen related receptors (ERRγ), is required to maintain mitochondrial function and drive the postnatal maturation of β cells. These observations emphasize that mitochondrial respiratory chain activities are also essential for beta-cell maturation. However, mitochondrial function and dysfunction have been implicated in many different aspects in the crosstalk with the nucleus, notably through ROS production and calcium signaling. Here, we have shown that p43 deletion induced oxidative stress in β cells detected two days after birth and that it was subsequently partly offset by an increase in the expression of the antioxidant enzymes in these cells. Since Kondo et al. showed that oxidative stress was responsible for a decrease in MafA stability via activation of p38 MAPK [42], we hoped to restore a glucose-stimulated insulin secretion with antioxidant supplementation in p43-/- mice. However, this was not the case. Perhaps the supplementation was a bit too drastic and inhibited normal favorable ROS signaling in β cells. This result also suggests that the retrograde signaling between the mitochondria and the nucleus induced by p43 and responsible for the regulation of MafA expression is not mediated by ROS.

Recently, the involvement of thyroid hormone in the functional maturation of the pancreatic beta-cells emerged. We reported that p43 depletion in mice induced a reduction of islet density and a loss of glucose-stimulated insulin secretion [20]. Aguayo-Mazzucato and coworkers have shown that TH drove the MafA expression through the direct binding of TRβ isoform on his promoter [8,10]. In addition, the authors found that TRα predominates at early ages in beta-cells, whereas TRβ becomes the predominant isoform in adult islets [10]. Moreover, Aguayo-Mazzucato et al. [33] have shown that MafA increases in expression in parallel to the acquisition of glucose responsiveness during the postnatal development before weaning and drove the beta-cell maturation. Altogether, these data indicate that in vivo, thyroid hormone regulates MafA expression in β cells differently depending on the temporal expression of TR isoforms. Before weaning, the TRα gene through the p43 and the regulation of mitochondrial activity induces the expression of MafA, whereas, in adult β cells, TRβ expression dominates and maintains the high levels of MafA. Lastly, during aging, TRα improves the expression of p16Ink4a, a marker and effector of senescence of β cells [10].

## 4. Materials and Methods

### 4.1. Animals and Ethics Statement

Male mice were housed and maintained on a 12-hour light/dark cycle (lights on at 7:30 pm). Food and water were provided *ad libitum*. Enrichment (nesting cotton squares) was provided in each cage. All animal experiments were performed according to European directives (86/609/CEE and 2010/63/CEE) and approved by the regional animal experimentation ethics committe: Région Languedoc-Roussillon (No36). Our institutional guidelines for the care and use of laboratory animals were observed. Our animal facility is approved by the Departmental Veterinary Services (No. E34-172-10, 2019/03/04) and the Ministry of Research (DUO No. 7053, 2020/02/26). P43-/- mice, lacking specifically the mitochondrial T3 receptor p43, were generated in our team as described previously [20]. All the mice used in these studies were back-crossed more than ten times into the C57BL/6 J background. According to the European Directive 210-63-EU, mice were observed daily for the general health status and mortality scoring.

### 4.2. Antioxidant Supplementation

For the antioxidant supplementation, coenzyme Q10 (0.25% *w/w*), alpha-lipoic acid (0.1% *w/w*) and vitamin E acetate (1000 IU/kg) were added to the standard rodent diet (SAFE-diets, Augy, France) by the manufacturer as previously described [43]. The combination of antioxidants was chosen to target different pathways of oxidative stress. Vitamin E is a well-known lipophilic-free radical scavenger that prevents ROS-induced cellular damage [44]. Coenzyme Q10 is a lipid-soluble antioxidant found in the mitochondria that have been shown to reduce lipid peroxidation in pancreatic mitochondria of diabetic rats in combination with vitamin E [45]. Alpha-lipoic acid is an amphiphilic antioxidant scavenger that has been shown to protect pancreatic β cells under oxidative stress conditions [46]. Female p43-/- mice received antioxidant supplementation 2 weeks before mating and until weaning.

### 4.3. Tissue Preparation, Laser Capture Microdissection and RNA Isolation

The pancreases were removed immediately after death and fixed in formalin for 4 hours at 4 °C. Each pancreas was sectioned (10 µm thickness) onto PET membrane glass slides (Leica, 11505151). Eosin-stained slides were placed on Leica LMD 7000 (Leica Microsystems, Nanterre, France). Illumination contrast and desired magnification objective were adjusted for optimal visualization. Using simple drawing tools, pancreatic islets were captured and transferred onto a 0.5 mL microcentrifuge tube cap containing 50 µL of extraction buffer. Total RNA was isolated using PicoPure RNA isolation kit according to the manufacturer’s protocol (Arcturus, KIT0204, Thermo Fisher Scientific, Waltham, MA, USA).

### 4.4. Gene Expression Analysis

Total RNA was then converted to complementary DNA (cDNA) using a prime script reagent kit (Takara Bio, Saint-Germain-en-Laye, France). Freshly synthesized cDNA then underwent preamplification (TATAA PreAMP GrandMaster Mix, 18 cycles) (TATAA Biocenter, Göteborg, Sweden) before qPCR. Quantitative PCR reactions were performed on the preamplified cDNA in the presence of Syber Green on a StepOnePlus real-time PCR system (Thermo Fisher Scientific, Waltham, MA, USA). Primer sequences are listed (Table 1).

### 4.5. Histological Analyses

Immunofluorescence and immunohistochemistry were performed as previously described [47,48]. Briefly, after antigen retrieval, 5 µm formalin-fixed pancreatic sections were incubated with the indicated antibodies overnight at 4 °C and with the indicated Alexa Fluor^TM^ secondary antibodies for one hour at room temperature (Thermo Fisher, Waltham, MA, USA). Nuclei were stained with Hoechst. Sections were mounted with PermaFluor (Thermo Fisher Scientific). The following antibodies were used: guinea pig anti-insulin (1/250; ab7842 Abcam, Cambridge, UK), mouse anti-glucagon (1/100; ab10988 Abcam), rabbit anti-MafA (1/100; IHC-00352 Bethyl Laboratories, Montgomery, TX, USA), mouse anti-PDX1 (1/200; F25A13B Developmental Studies Hybridoma Bank at the University of Iowa (Iowa City, IA, USA)), rabbit anti-PH3 (1/500; 06-570 Millipore, Merck KGaA, Darmstadt, Germany), rabbit anti-HNE (1/100; ab 46545 Abcam).

### 4.6. Quantification

For the analyses, pancreatic sections with immunofluorescence staining were scanned using an AxioScan.Z1 (Zeiss, Marly-le-Roi, France) with a 20X objective, whereas pancreatic sections with hematoxylin–eosin staining were scanned using a NanoZoomer (Hamamatsu Photonics, Massy, France) with a 20X objective. The surface of islets was assessed on hematoxylin–eosin sections by NDP.View 2 software (Hamamatsu Photonics, Massy, France). The total number of immune-positive cells by islet and staining intensity was quantified by ZEN 2.3 software (Zeiss, Marly-le-Roi, France). The intensity of fluorescence in the beta-cells was quantified using the ZEN 2.3 software following the next procedure. After a z-projection by summing slice intensity of all stacks containing the pancreas section, regions of interest (ROI) of each islet and of a large area dedicated to background measurement were drawn. Measurements of the area and the mean intensity were done for each ROI. The fluorescent background was then subtracted from the measure of the total intensity in beta-cells using this formula = ((mean of fluorescence intensity measured) − (mean of fluorescence intensity measured for the background)) × area of the ROI). The rate of beta-cell proliferation was measured by counting the number of PH3 and insulin-positive cells over the total number of insulin-expressing cells.

### 4.7. Insulin Concentration

Insulin was measured in plasma using an Ultra Sensitive Mouse Elisa kit as described by the manufacturer (Crystal Chem, Zaandam, The Netherlands).

### 4.8. Statistical Analyses

All results are presented as means ± SEM or as percentages. Statistical significances of the differences between groups were evaluated with the Student’s t-test.

## 5. Conclusions

In conclusion, we have shown that p43, a mitochondrial T3 receptor, drives beta-cell maturation in the postnatal period via induction of transcription factor MafA. In addition, if antioxidant supplementation restores a normal islet density in p43-/- mice, it fails to restore a glucose-stimulated insulin secretion suggesting that the retrograde signaling between the mitochondria and the nucleus responsible for the regulation of MafA expression is not mediated by ROS.

Altogether, these data indicate that in vivo, thyroid hormone regulates MafA expression and maturation of β cells differently depending on the temporal expression of TR isoforms. Before weaning, the TRα gene through the p43 and the regulation of mitochondrial activity induces the expression of MafA, whereas, in adult β cells, TRβ expression dominates and maintains the high levels of MafA.

## Figures and Tables

**Figure 1 ijms-22-02489-f001:**
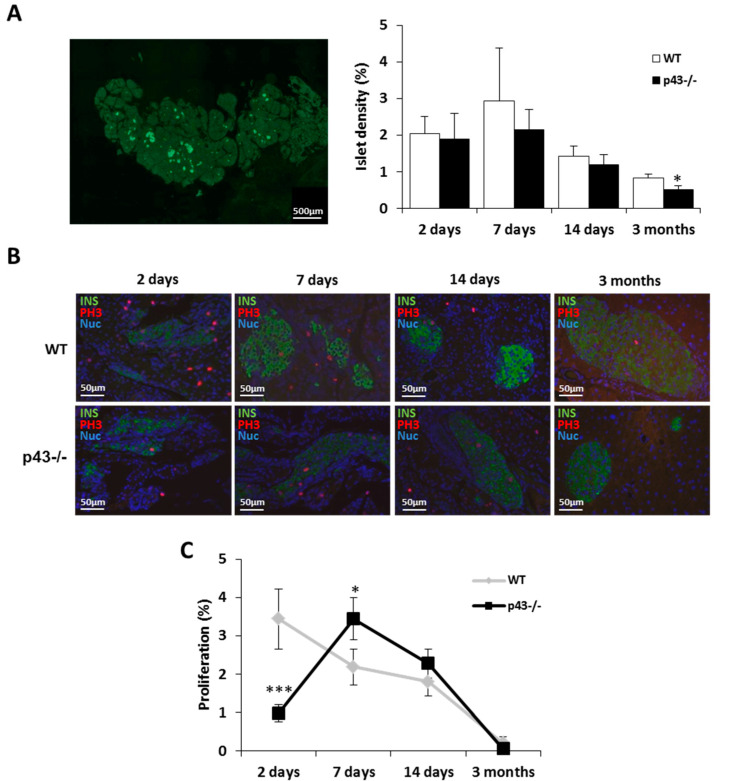
Beta-cell proliferation is affected in mice lacking p43. (**A**) Picture of insulin content (green) in pancreatic section and quantification of pancreatic islets density from WT and p43-/- mice at 2 days, 7 days, 14 days and 3 months after birth (*n* = 5 animals for each age and each genotype). (**B**) Representative pictures of replicating beta-cells (insulin (green), PH3 (red) and Hoescht (blue)) in pancreatic sections from WT and p43-/- mice at 2 days, 7 days, 14 days and 3 months after birth. (**C**) Quantification of replicating beta-cells by islet (%) in pancreatic sections from WT and p43-/- mice at 2 days, 7 days, 14 days and 3 months after birth (*n* = 5 animals for each age and each genotype). Statistical significance: * *p* < 0.05; *** *p* < 0.001, Student’s t-test. Results are expressed as mean ± SEM. INS: insulin, PH3: phosphohistone H3, Nuc: nucleus.

**Figure 2 ijms-22-02489-f002:**
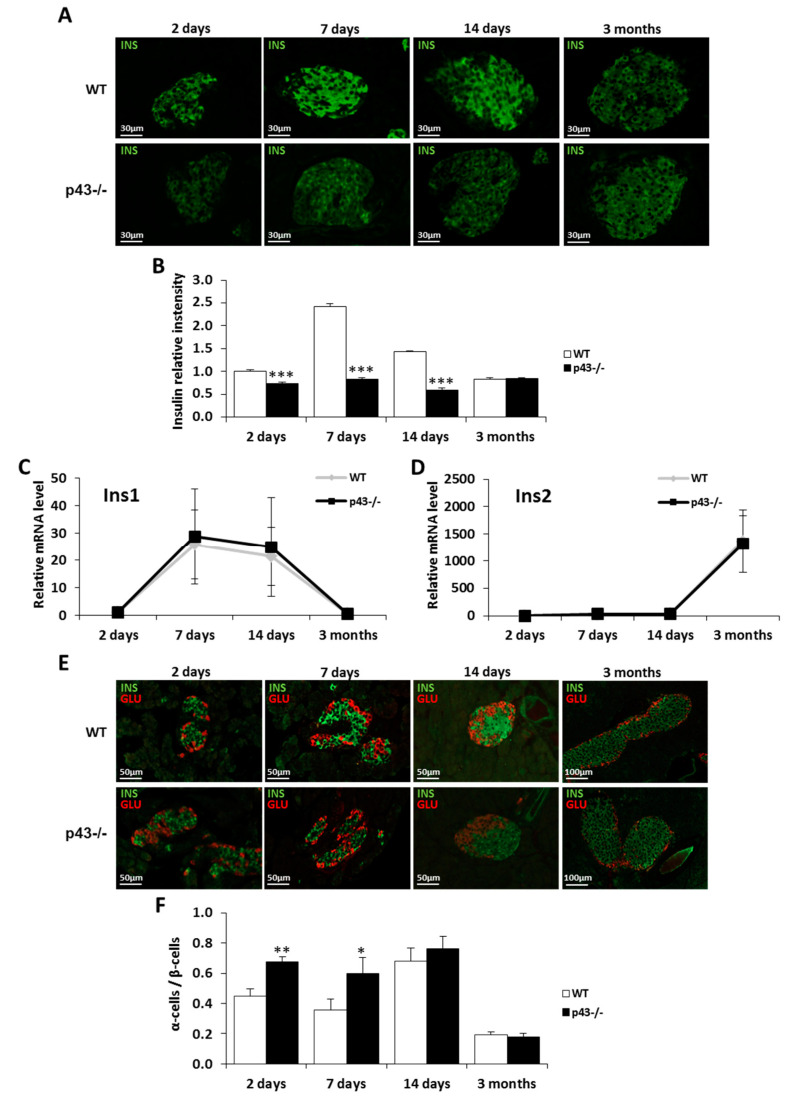
Deletion of p43 decreases insulin content and affects pancreatic islet composition in the postnatal period. (**A**) Representative pictures of insulin content (green) in pancreatic sections from WT and p43-/- mice at 2 days, 7 days, 14 days and 3 months after birth. (**B**) Quantification of relative insulin intensity by islet (%) in pancreatic sections from WT and p43-/- mice at 2 days, 7 days, 14 days and 3 months after birth (*n* = 5 animals for each age and each genotype). (**C**,**D**) Relative mRNA expression of Ins1 and Ins2 in WT and p43-/- mice at 2 days, 7 days, 14 days and 3 months after birth (*n* = 5 for each age and each genotype). (**E**) Representative pictures of the overall distribution of alpha (glucagon, red) and beta-cells (insulin, green) in pancreatic sections from WT and p43-/- mice at 2 days, 7 days, 14 days and 3 months after birth. (**F**) Quantification of alpha cell/beta-cell ratio in pancreatic sections from WT and p43-/- mice at 2 days, 7 days, 14 days and 3 months after birth (*n* = 5 animals for each age and each genotype). Statistical significance: * *p* < 0.05; ** *p* < 0.01; *** *p* < 0.001, Student’s *t*-test. Results are expressed as mean ± SEM. INS: insulin, GLU: glucagon.

**Figure 3 ijms-22-02489-f003:**
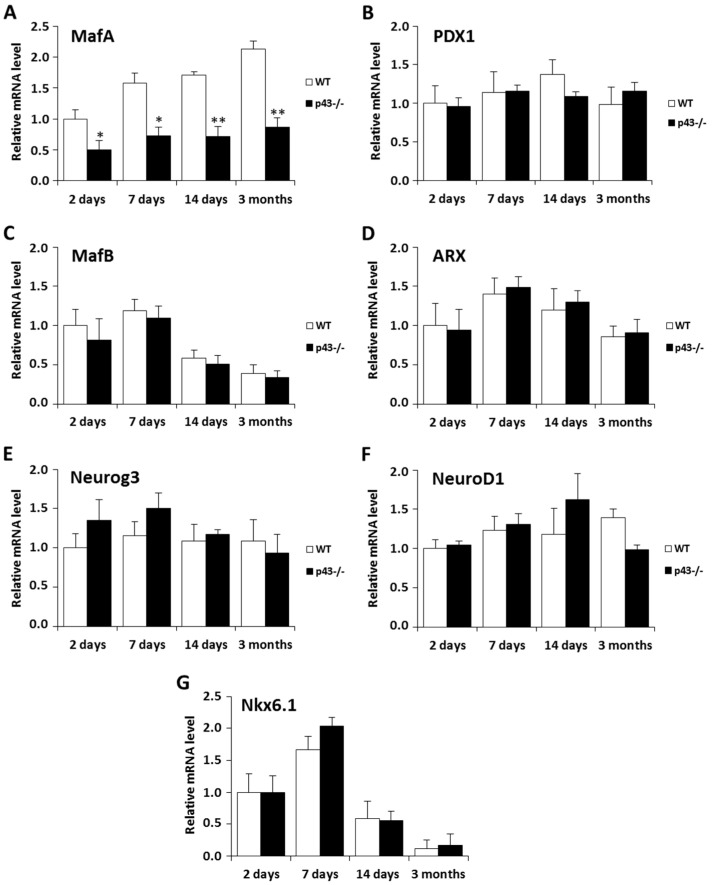
MafA expression at mRNA level is strongly decreased in mice lacking p43. (**A–G**) Relative mRNA expression of islet transcription factors in WT and p43-/- mice at 2 days, 7 days, 14 days and 3 months after birth (*n* = 5 for each age and each genotype). Statistical significance: * *p* < 0.05; ** *p* < 0.01, Student’s *t*-test. Results are expressed as mean ± SEM.

**Figure 4 ijms-22-02489-f004:**
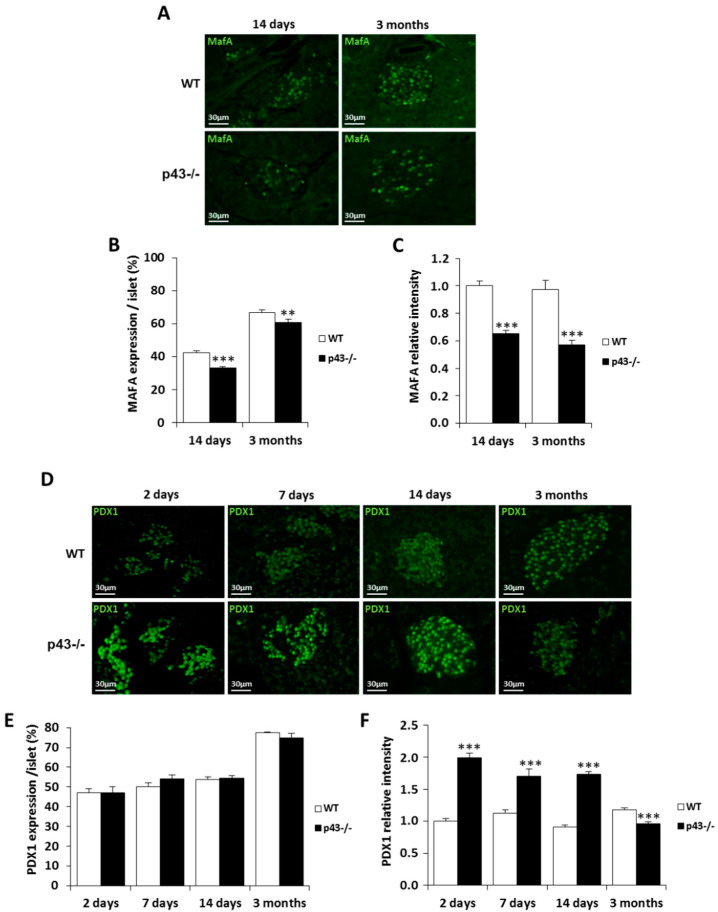
MafA expression is strongly decreased in mice lacking p43. (**A**) Representative pictures of MafA expression (green) in pancreatic sections from WT and p43-/- mice at 2 days, 7 days, 14 days and 3 months after birth. (**B**) Quantification of beta-cells expressing MafA by islet (%) in pancreatic sections from WT and p43-/- mice at 2 days, 7 days, 14 days and 3 months after birth (*n* = 5 animals for each age and each genotype). (**C**) Quantification of relative MafA intensity by islet (%) in pancreatic sections from WT and p43-/- mice at 2 days, 7 days, 14 days and 3 months after birth (*n* = 5 animals for each age and each genotype). (**D**) Representative pictures of PDX1 expression (green) in pancreatic sections from WT and p43-/- mice at 2 days, 7 days, 14 days and 3 months after birth. (**E**) Quantification of beta-cells expressing PDX1 by islet (%) in pancreatic sections from WT and p43-/- mice at 2 days, 7 days, 14 days and 3 months after birth (*n* = 5 animals for each age and each genotype). (**F**) Quantification of relative PDX1 intensity by islet (%) in pancreatic sections from WT and p43-/- mice at 2 days, 7 days, 14 days and 3 months after birth (*n* = 5 animals for each age and each genotype). Statistical significance: ** *p* < 0.01; *** *p* < 0.001, Student’s *t*-test. Results are expressed as mean ± SEM.

**Figure 5 ijms-22-02489-f005:**
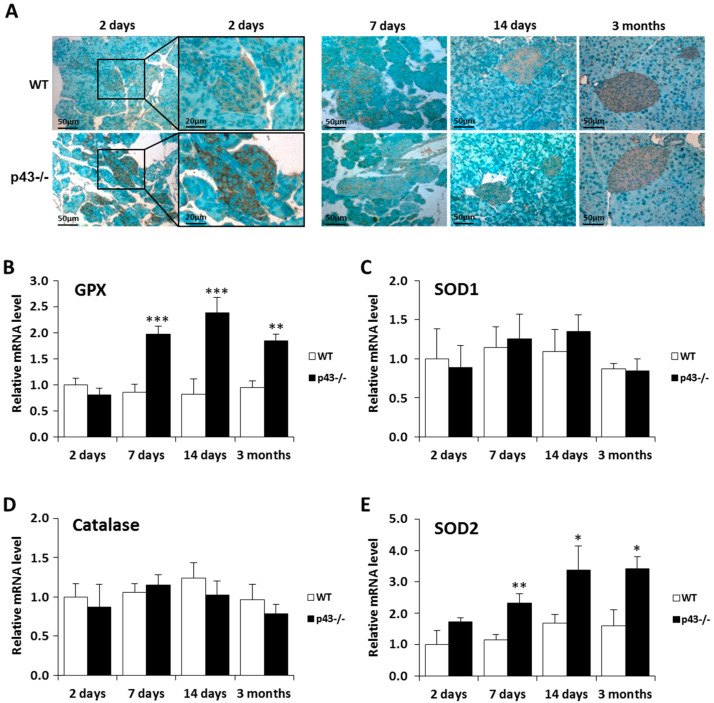
Deletion of p43 induces oxidative stress in pancreatic islet. (**A**) Representative pictures of HNE staining in pancreatic sections from WT and p43-/- mice at 2 days, 7 days, 14 days and 3 months after birth. (**B**–**E**) Relative mRNA expression of antioxidant enzymes in WT and p43-/- mice at 2 days, 7 days, 14 days and 3 months after birth (*n* = 5 for each age and each genotype). Statistical significance: **p* < 0.05; ***p* < 0.01; ****p* < 0.001, Student’s *t*-test. Results are expressed as mean ± SEM.

**Figure 6 ijms-22-02489-f006:**
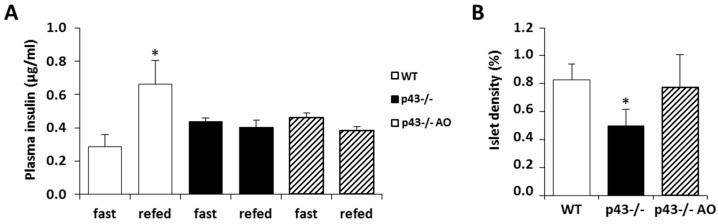
Antioxidant supplementation in pregnant females partly restores pancreatic islet phenotype in p43-/- mice. (**A**) Plasma insulin levels were recorded in fasting condition or 30 minutes after a glucose administration (2 g/kg) by oral gavage. Experiments were performed at 3 months of age in WT, p43-/- and p43-/- mice supplemented (p43-/- AO) with antioxidants (*n* = 7 for each group). (**B**) Quantification of pancreatic islets density from WT, p43-/- and p43-/- AO mice at 3 months of age (*n* = 7 animals for each group). Statistical significance: * *p* < 0.05, Student’s *t*-test. Results are expressed as mean ± SEM.

**Table 1 ijms-22-02489-t001:** Primers list.

mMafA	Foward	AGGCCTTCCGGGGTCAGAG
	Reverse	TGGAGCTGGCACTTCTCGCT
mMafb	Foward	GCAGGTATAAACGCGTCCAG
	Reverse	TGAATGAGCTGCGTCTTCTC
mPdx1	Foward	CTTAACCTAGGCGTCGCACAA
	Reverse	GAAGCTCAGGGCTGTTTTTCC
mNeuroD1	Foward	CTTGGCCAAGAACTACATCTGG
	Reverse	GGAGTAGGGATGCACCGGGAA
mNeurog3	Foward	AAACTCCAAAGGGTGGATGA
	Reverse	TGTGCCAGCCTCTGACTTAG
mArx	Foward	TTCCAGAAGACGCACTACCC
	Reverse	TCTGTCAGGTCCAGCCTCAT
mNkx6.1	Foward	CCTTAGTATCCCTGCCTTCTCTC
	Reverse	AGAGGACCGACGGCTGTT
mIns1	Foward	CAGAGAGGAGGTACTTTGGACTATAAA
	Reverse	GCCATGTTGAAACAATGACCT
mIns2	Foward	GAAGTGGAGGACCCACAAGT
	Reverse	AGTGCCAAGGTCTGAAGGTC
mSOD1	Foward	TGAGGTCCTGCACTGGTAC
	Reverse	CAAGCGGTGAACCAGTTGTG
mSOD2	Foward	ATCTGTAAGCGACCTTGCTC
	Reverse	GCCTGCACTGAAGTTCAATG
mGPX1	Foward	TTCCGCAGGAAGGTAAACAGC
	Reverse	GTCTCTCTGAGGCACGATCCG
mCatalase	Foward	GCATGCACATGGGGCCATCA
	Reverse	ACCCTCTTATACCAGTTGGC

## Data Availability

The data presented in this study are available on request from the corresponding author.

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
