# Peer review of "p43, a Truncated Form of Thyroid Hormone Receptor α, Regulates Maturation of Pancreatic β Cells"

_ijms, 2021, doi:10.3390/ijms22052489_

Round 1
Reviewer 1 Report
This study was designed to determine whether p43 is involved in the development and maturation of β cells. The authors demonstrated that p43 drives the maturation of β cells by induction of the transcription factor MafA during the postnatal period. The results of the study will make a worthwhile contribution to the literature on this subject.
I have the following questions and suggestions for changes in the manuscript.
line 15: Change to "analysis".
lines 18 and 257: Change to "Lastly".
line 37: Change to "drives". Should this be "MafA"?
line 43: Change to "mass".
line 57: Change to "Because the postnatal period is a critical window...".
line 63: Change to "...3-month old mice...".
line 71: Change to "neonatal mice".
line 79: Change to "...n=5 animals for each age and each genotype...".
lines 80, 106, 119, 151, 173, and 193: I assume this should read "Results are expressed as mean ± SEM".
line 82: Change to "evaluated".
line 87: I am not sure what you mean by "remained constant when it peaked at 7 days". Remained constant from when to when?
line 93: I do not understand what you mean by "to reach then adult values".
line 95: Change "no" to "not".
Figure 2B: Insulin is misspelled on the Y-axis.
Figure 3A: The time labels are missing on the X-axis.
lines 124 and 133: Change to "evaluated".
line 132: Change to "increased".
line 136: Change to "...slightly decreased in adults...".
line 154: Change "increase" to "increases".
line 158: Change "were" to "was". Here and throughout the manuscript, make sure that the subject and verb agree.
line 179: "plasma insulin levels were measured in plasma". It is redundant to say "plasma" twice.
line 186: Change to "...partially restores normal pancreatic β cell phenotypes".
line 200: Change to "...during the post-natal period, which cannot explain...".
lines 200 and 215: "post-natal" or "postnatal"?
line 217: Change to "...contrast to a previous publication...".
line 218: Change to "observed".
lines 224-226: Change to "Because overexpression of PDX1 is also known to increase insulin content [33,31] we postulate that this increase in PDX1 is most likely an adaptation to compensate in part for the decrease in insulin synthesis in beta cells in response to the fall of MafA expression".
line 228: Change to "This set of data demonstrates that...".
line 249: Change "his" to "its".
line 253: "drives" in place of "drove".
line 263: Change to "...and during the lifespan [38]".
lines 271 and 357: Change to "institutional".
line 280: Change to "...added to a standard rodent diet...".
line 282: Change to "Vitamin E is a well known...".
line 289: The meaning of "fixed in formalin 4 hours" is unclear. Please reword.
line 293: "transferred onto 0.5 mL micro-centrifuge tube cap". Transferred onto the cap?
line 329: "Area" does not need to be capitalized.
line 334: insert "the" before "manufacturer".
line 337: insert "the" before "Student’s t-test".
lines 339-340: Change to "...drives the maturation of beta cells in the postnatal period...".
lines 341-342: Change to "it fails to restore glucose-stimulated insulin secretion, suggesting...".
Author Response
Reviewer 1:
This study was designed to determine whether p43 is involved in the development and maturation of β cells. The authors demonstrated that p43 drives the maturation of β cells by induction of the transcription factor MafA during the postnatal period. The results of the study will make a worthwhile contribution to the literature on this subject.
Response: I am very grateful to the reviewer for his comments and for the corrections made to the manuscript.
I have the following questions and suggestions for changes in the manuscript.
line 15: Change to "analysis". OK
lines 18 and 257: Change to "Lastly". OK
line 37: Change to "drives". Should this be "MafA"? OK
line 43: Change to "mass". OK
line 57: Change to "Because the postnatal period is a critical window...". OK
line 63: Change to "...3-month old mice...". OK
line 71: Change to "neonatal mice". OK
line 79: Change to "...n=5 animals for each age and each genotype...". OK
lines 80, 106, 119, 151, 173, and 193: I assume this should read "Results are expressed as mean ± SEM". OK
line 82: Change to "evaluated". OK
line 87: I am not sure what you mean by "remained constant when it peaked at 7 days". Remained constant from when to when? It was deleted.
line 93: I do not understand what you mean by "to reach then adult values". OK
line 95: Change "no" to "not". OK
Figure 2B: Insulin is misspelled on the Y-axis. OK
Figure 3A: The time labels are missing on the X-axis. OK
lines 124 and 133: Change to "evaluated". OK
line 132: Change to "increased". OK
line 136: Change to "...slightly decreased in adults...". OK
line 154: Change "increase" to "increases". OK
line 158: Change "were" to "was". Here and throughout the manuscript, make sure that the subject and verb agree. OK
line 179: "plasma insulin levels were measured in plasma". It is redundant to say "plasma" twice. OK
line 186: Change to "...partially restores normal pancreatic β cell phenotypes". OK
line 200: Change to "...during the post-natal period, which cannot explain...". OK
lines 200 and 215: "post-natal" or "postnatal"? OK
line 217: Change to "...contrast to a previous publication...". OK
line 218: Change to "observed". OK
lines 224-226: Change to "Because overexpression of PDX1 is also known to increase insulin content [33,31] we postulate that this increase in PDX1 is most likely an adaptation to compensate in part for the decrease in insulin synthesis in beta cells in response to the fall of MafA expression". OK
line 228: Change to "This set of data demonstrates that...". OK
line 249: Change "his" to "its". OK
line 253: "drives" in place of "drove". OK
line 263: Change to "...and during the lifespan [38]". OK
lines 271 and 357: Change to "institutional". OK
line 280: Change to "...added to a standard rodent diet...". OK
line 282: Change to "Vitamin E is a well known...". OK
line 289: The meaning of "fixed in formalin 4 hours" is unclear. Please reword. OK
line 293: "transferred onto 0.5 mL micro-centrifuge tube cap". Transferred onto the cap? Yes.
line 329: "Area" does not need to be capitalized. OK
line 334: insert "the" before "manufacturer". OK
line 337: insert "the" before "Student’s t-test". OK
lines 339-340: Change to "...drives the maturation of beta cells in the postnatal period...". OK
lines 341-342: Change to "it fails to restore glucose-stimulated insulin secretion, suggesting...". OK
Reviewer 2 Report
This study aims to demonstrate that p43 loss negatively affects beta cell maturation through changes to ROS and MafA gene expression. While interesting, the reviewer is left with many concerns that will need to be addressed with additional experiments.
Figure 1
-Why are only male animals used? Clear affects of sex differences abound in biology. This study should really look at female and male animals.
-There appears to be a disconnect between the change in percent proliferation and islet density (more appropriately percent islet area). No change in density is observed until 3 months, however the impaired proliferation is observed between 2 and 7 days (or earlier? in utero levels should be measured). This leads to two questions-1) how does the number of beta cells change, 2) how does the beta cell size change (not total cumulative beta cell area, but the actual perimeter or volume of the beta cells).
-PH3 is a great marker, for looking at mitosis. In turns of proliferation it is only half the story. DNA replication should really be measured, as PH3 positive cells may get stuck and give an inflated level or cell cycle progression. BrdU or EdU incorporation should be presented, or at the very least Ki67. With these two markers you can define DNA replication and mitotic entry, defining true proliferation.
Figure 2
-Relative fluorescent intensity is a good starting point, but this must be validated by other means. If insulin content is presumably down, then one should see decreased total insulin in isolated islets. This is a straight forward process and should be presented. Similarly, if the protein is down, is this due to transcription or processing? At the very least insulin mRNA should be measured.
-Again, there appears to be a disconnect between the alpha/beta cell ratio in 2D and the data in 1A. How can islet density (really beta cell area) be decreased at 3 months, with no change in alpha to beta cell ratio at 3 months? Either the calculations are wrong, or your alpha cell numbers are decreasing. The 2 day and 7 day data is also unclear. Is this elevated in the KO because there are more alpha cells or less beta cells? If it is less beta cells, how does this fit with the data in 1A?
Figure 3
-The decreased expression of MafA is interesting. Do insulin expression change similarly? What about Nkx6.1? What are the phenotypic changes that should manifest with decreased MafA levels? Do they change? And....
Figure 4
-....your MafA IF data (and mRNA data) has to be validated with western blotting. If your claim is that p43 KO decreases MafA, then protein should be down in isolated islet westerns.
-Same is true for Pdx1, absolutely need to show western blotting validation.
-Does p43 ChIP to the MafA promoter? Is there a p43 binding site in the MafA promoter?
Figure 5
-Need quantitation for the data in 5A
-What is the mechanism for increasing GPX and SOD2? How is p43 doing this? Does p43 act as a repressor, and then its loss leads to increased expression? This needs to be addressed.
Figure 6
-This figure doesn't fit. It seems to just be thrown in here. It is also unclear regarding the stats in 6B. Is the significance of p43-/- vs WT? Is the difference between p43-/- and p43-/-AO significant? What is the mechanism?
In addition there are many grammatical errors in the text that need to be fixed (As an example, line 95, although this is not the only error).
Author Response
Reviewer 2:
This study aims to demonstrate that p43 loss negatively affects beta cell maturation through changes to ROS and MafA gene expression. While interesting, the reviewer is left with many concerns that will need to be addressed with additional experiments.
Response: I am very grateful to the reviewer for his comments.
Figure 1
-Why are only male animals used? Clear affects of sex differences abound in biology. This study should really look at female and male animals.
Response: I agree with the reviewer that the ideal is to experiment on males and females. However, the first studies showing the role of p43 on carbohydrate homeostasis have only been done in male mice. Moreover, the work on animals of both sexes also raises the problem of the need for additional animals, which goes against our ethics committee's request to use as few animals as possible.
-There appears to be a disconnect between the change in percent proliferation and islet density (more appropriately percent islet area). No change in density is observed until 3 months, however the impaired proliferation is observed between 2 and 7 days (or earlier? in utero levels should be measured). This leads to two questions-1) how does the number of beta cells change, 2) how does the beta cell size change (not total cumulative beta cell area, but the actual perimeter or volume of the beta cells).
Response: As mentioned in discussion part, the deletion of p43 affects only slightly proliferation of β cells during the post-natal period which cannot explain the decrease in islet density observed in adult mice. These data indicate that the mechanisms involved in the reduction of islet density in adulthood in p43-/- mice occur more probably after the neonatal period studied here. It is possible that the period around weaning could be important but it has not been tested in this study.
-PH3 is a great marker, for looking at mitosis. In turns of proliferation it is only half the story. DNA replication should really be measured, as PH3 positive cells may get stuck and give an inflated level or cell cycle progression. BrdU or EdU incorporation should be presented, or at the very least Ki67. With these two markers you can define DNA replication and mitotic entry, defining true proliferation.
Response: We also tested Ki67 and had results equivalent to those obtained with PH3. However, immunofluorescence staining was more intense with PH3 than with Ki67.
Figure 2
-Relative fluorescent intensity is a good starting point, but this must be validated by other means. If insulin content is presumably down, then one should see decreased total insulin in isolated islets. This is a straight forward process and should be presented. Similarly, if the protein is down, is this due to transcription or processing? At the very least insulin mRNA should be measured.
Response: In this study it is not realistic to make isolated islets to measure insulin content. On 2 and 7 day old mice it would take at least 5 mice to make 1 sample and at least 2 mice at 15 days and 3 months. Knowing that at least 5 samples are needed to make statistics this would represent a very large number of mice to produce (70 p43-/- male mice and 70 WT male mice). This is not possible from an ethical point of view.
However, we quantified by QPCR the expression of Ins1 and Ins2 (the 2 genes coding for insulin in mice) from RNA extracted from islets isolated by laser microdissection. We saw no difference in the expression of these genes in the absence of p43 (Figure 2C-D). These results suggest that the decrease in insulin content observed in the beta cells of p43-/- mice may be related to a decrease in mRNA translation.
-Again, there appears to be a disconnect between the alpha/beta cell ratio in 2D and the data in 1A. How can islet density (really beta cell area) be decreased at 3 months, with no change in alpha to beta cell ratio at 3 months? Either the calculations are wrong, or your alpha cell numbers are decreasing. The 2 day and 7 day data is also unclear. Is this elevated in the KO because there are more alpha cells or less beta cells? If it is less beta cells, how does this fit with the data in 1A?
Response: Our results confirmed that in adult p43-/- mice there was no change in the alpha/beta cell ratio as previously shown (Blanchet et al, 2012; Bertrand et al, 2013). This indicates that the number of alpha cells is also decreased in p43-/- mice.
Figure 3
-The decreased expression of MafA is interesting. Do insulin expression change similarly? What about Nkx6.1? What are the phenotypic changes that should manifest with decreased MafA levels? Do they change? And....
Response: at mRNA level Ins1 and Ins2 expression was not affected by p43 deletion (Fig 2C-D). Nkx6.1 level was not modified in p43-/- mice (Fig 3G).
Figure 4
-....your MafA IF data (and mRNA data) has to be validated with western blotting. If your claim is that p43 KO decreases MafA, then protein should be down in isolated islet westerns.
Response: IF quantification is recognized as quantitative. Western-blot would also have allowed to confirm this MafA decrease but as previously indicated it is not realistic to make isolated islets to perform such experiments. On 2 and 7 day old mice it would take at least 5 mice to make 1 sample and at least 2 mice at 15 days and 3 months. Knowing that at least 5 samples are needed to make statistics this would represent a very large number of mice to produce (70 p43-/- male mice and 70 WT male mice). This is not possible from an ethical point of view.
-Same is true for Pdx1, absolutely need to show western blotting validation.
Response: see above
-Does p43 ChIP to the MafA promoter? Is there a p43 binding site in the MafA promoter?
Response: P43 is a mitochondrial transcription factor. It is not localized in the nucleus and therefore cannot bind to the MafA gene promoter.
Figure 5
-Need quantitation for the data in 5A
Response: It is very difficult to reliably quantify the peroxidase stains we performed with the anti-HNE antibody because we have a methylene green counter-stain. However, the 2-day images indicate a strong oxidative stress in the islets of p43-/- mice compared to WT animals.
-What is the mechanism for increasing GPX and SOD2? How is p43 doing this? Does p43 act as a repressor, and then its loss leads to increased expression? This needs to be addressed.
Response: I think this is not a direct effect of p43 deletion. The increase in GPX and SOD2 expression is probably an adaptation of the islets to limit the harmful effects of oxidative stress induced by the absence of p43 otherwise we would probably have seen it at 2 days, which is not the case.
Figure 6
-This figure doesn't fit. It seems to just be thrown in here. It is also unclear regarding the stats in 6B. Is the significance of p43-/- vs WT? Is the difference between p43-/- and p43-/-AO significant? What is the mechanism?
Response: This experiment was designed to study the implication of ROS in the reduction of islet density and in a loss of glucose-stimulated insulin secretion in p43-/- mice. We observed that at the age of 3 months the offspring of female p43-/- mice, having received a combination of antioxidants until weaning, had no further decrease in islet density. As shown in Figure 6, the difference between p43-/- and WT is significant (p<0.05). The difference between p43-/- and p43-/-AO is sub-significant (p<0.08) and for this reason there is no asterix.
The mechanism by which p43 regulates nuclear gene expression is still unclear. However, it can be assumed that p43 as a mitochondrial transcription factor that modulates the activity of the respiratory chain must regulate the expression of many genes, notably through ROS production and calcium signaling.
In addition there are many grammatical errors in the text that need to be fixed (As an example, line 95, although this is not the only error).
Response: many corrections have been made.
Reviewer 3 Report
The work presented by Emilie Blanchet et al., entitled "p43, a truncated form of thyroid hormone receptor α, regulates maturation of pancreatic β cells," reported an in vivo study analyzing the effects of the truncated form of TRβ on the maturation and development of pancreatic β cells in mice.
Data were well presented; experimental procedures were well performed.
Minor revisions:
1)Fig. 1C: I suggest to the authors to represent data of the percentage of proliferation with a linear graph instead of the histogram to highlight the differences between the mice's two genotypes in terms of proliferation.
2)Fig. 3A: the name samples in the histogram are represented in white color instead of black like appear in the figure's other histograms.
Author Response
Reviewer 3:
The work presented by Emilie Blanchet et al., entitled "p43, a truncated form of thyroid hormone receptor α, regulates maturation of pancreatic β cells," reported an in vivo study analyzing the effects of the truncated form of TRβ on the maturation and development of pancreatic β cells in mice.
Data were well presented; experimental procedures were well performed.
Response: I am very grateful to the reviewer for his comments.
Minor revisions:
1)Fig. 1C: I suggest to the authors to represent data of the percentage of proliferation with a linear graph instead of the histogram to highlight the differences between the mice's two genotypes in terms of proliferation.
Response: The proliferation is now presented as a linear graph as suggested.
2)Fig. 3A: the name samples in the histogram are represented in white color instead of black like appear in the figure's other histograms.
Response: This was corrected.
Reviewer 4 Report
The authors have previously showed that deletion of p43 led to reduction of pancreatic islet density and a loss of glucose-stimulated insulin secretion in adult mice. The present study was designed to determine whether p43 was involved in the processes of β cells development and maturation. Neonatal, juvenile and adult p43-/- mice were used to analyze the development of β cells in the pancreas. Their findings demonstrated that p43 drives the β cells maturation via its induction of transcription factor MafA during the post-natal critical window.
Introduction
In this section thyroid hormone is introduced as a major regulator of metabolism and mitochondrial function, and also a key regulator of postnatal maturation of many tissues. It explains how it affects different metabolic aspects of glucose and insulin metabolism. They also introduce the different receptors involved and describe how they are associated with the pancreas islet and its beta cells.
Previously, they have identified a truncated form of the nuclear receptor TRα1, with molecular masses of 43 kDa (p43), which stimulates mitochondrial transcription and protein synthesis in the presence of T3. The aim of the present study was to determine whether p43 was involved in the processes of β cells development and maturation.
Results
They demonstrate that p43-/- deletion affects slightly the proliferation of β cells in neonate mice. This is nicely demonstrated in figure 1.
They next wanted to evaluate the influence of p43 deletion on insulin content by measuring the immunofluorescent labeling intensity using an antibody raised against insulin. Quantitative results showed a strong decrease of insulin content in p43-/- β cells in comparison to wild-type pancreas at 2, 7 and 14 days, whereas the content appeared normal in adult pancreas. They also evaluated the ratio of alfa- beta- cells. They demonstrated that p43-/- deletion affects insulin content and α/β ratio during the post-natal period but no in adult mouse. Well illustrated in figure 2.
They further showed that deletion of p43 strongly decreases MafA expression. They found that MafA mRNA expression remained dramatically low in p43-/- islet whereas it gradually increased in controls. In contrast, the mRNA expression of PDX1, MafB, ARX, Neurogenin 3 and NeuroD1 was not altered by p43 deletion.
MafA is introduced and explained in the introduction part, but PDX1, ARX, Neurogenin3 and NeuroD1 needs to be explained.
They further evaluated the effect of oxidative stress and antioxidant treatment in p43-/- mice.
Conclusively they show that the islet density in p43-/- neonates was similar to the controls. In addition, the deletion of p43 affects only slightly proliferation of β cells during the post-natal which cannot explain the decrease in islet density observed in adult mice. These data indicate that the mechanisms involved in the reduction of islet density in adulthood in p43-/- mice occur after the neonatal period. Moreover, the finding that antioxidant supplementation during pregnancy until weaning restores a normal islet density in p43-/- mice suggests that the proliferation defect is at least in part mediated by the activation of ROS production. They have further evaluated genes characteristic of mature functional beta cells.
All in all the discussion part is too long. It would preferably be better if it was more precise. Manuscript should be accepted with minor revision.
Author Response
Reviewer 4:
The authors have previously showed that deletion of p43 led to reduction of pancreatic islet density and a loss of glucose-stimulated insulin secretion in adult mice. The present study was designed to determine whether p43 was involved in the processes of β cells development and maturation. Neonatal, juvenile and adult p43-/- mice were used to analyze the development of β cells in the pancreas. Their findings demonstrated that p43 drives the β cells maturation via its induction of transcription factor MafA during the post-natal critical window.
Introduction
In this section thyroid hormone is introduced as a major regulator of metabolism and mitochondrial function, and also a key regulator of postnatal maturation of many tissues. It explains how it affects different metabolic aspects of glucose and insulin metabolism. They also introduce the different receptors involved and describe how they are associated with the pancreas islet and its beta cells.
Previously, they have identified a truncated form of the nuclear receptor TRα1, with molecular masses of 43 kDa (p43), which stimulates mitochondrial transcription and protein synthesis in the presence of T3. The aim of the present study was to determine whether p43 was involved in the processes of β cells development and maturation.
Results
They demonstrate that p43-/- deletion affects slightly the proliferation of β cells in neonate mice. This is nicely demonstrated in figure 1.
They next wanted to evaluate the influence of p43 deletion on insulin content by measuring the immunofluorescent labeling intensity using an antibody raised against insulin. Quantitative results showed a strong decrease of insulin content in p43-/- β cells in comparison to wild-type pancreas at 2, 7 and 14 days, whereas the content appeared normal in adult pancreas. They also evaluated the ratio of alfa- beta- cells. They demonstrated that p43-/- deletion affects insulin content and α/β ratio during the post-natal period but no in adult mouse. Well illustrated in figure 2.
They further showed that deletion of p43 strongly decreases MafA expression. They found that MafA mRNA expression remained dramatically low in p43-/- islet whereas it gradually increased in controls. In contrast, the mRNA expression of PDX1, MafB, ARX, Neurogenin 3 and NeuroD1 was not altered by p43 deletion.
MafA is introduced and explained in the introduction part, but PDX1, ARX, Neurogenin3 and NeuroD1 needs to be explained.
Response: PDX1, ARX, Neurogenin3, NeuroD1 and Knx6.1 are presented in the result section.
They further evaluated the effect of oxidative stress and antioxidant treatment in p43-/- mice.
Conclusively they show that the islet density in p43-/- neonates was similar to the controls. In addition, the deletion of p43 affects only slightly proliferation of β cells during the post-natal which cannot explain the decrease in islet density observed in adult mice. These data indicate that the mechanisms involved in the reduction of islet density in adulthood in p43-/- mice occur after the neonatal period. Moreover, the finding that antioxidant supplementation during pregnancy until weaning restores a normal islet density in p43-/- mice suggests that the proliferation defect is at least in part mediated by the activation of ROS production. They have further evaluated genes characteristic of mature functional beta cells.
All in all the discussion part is too long. It would preferably be better if it was more precise. Manuscript should be accepted with minor revision.
Response: I am very grateful to the reviewer for this comment. The discussion section was slightly shortened as suggested.
Round 2
Reviewer 2 Report
The questions raised in the original submission, regarding the need to validate changes using other molecular biology techniques (western blotting of isolated islets, ChIP at the promoters of genes with changed expression, etc.) are still essential. Increasing the number of animals is acceptable if needed to accurately answer the scientific questions. Or as an alternative approach p43 could be knocked out in a beta cell line to address biochemically the questions in a similar way. As the authors pointed out that p43 is a mitochondrial TF, an explanation should be given for how changes in p43 expression affect nuclear gene expression of MafA (and frankly without some protein measurement this change means little). The questions previously raised need to be addressed with additional experimentation.
Author Response
How changes in p43 expression affect nuclear gene expression? This is a recurring question that I would like to answer and that arises whenever studies are conducted on the impact of changes in mitochondrial activity on nuclear gene expression.
Although mitochondria have their own DNA that codes for 13 respiratory chain proteins, the vast majority (>99%) of mitochondrial proteins are encoded in the nucleus. Communication between the mitochondria and the nucleus is therefore necessary not only to coordinate the synthesis of mitochondrial proteins during biogenesis, but also to communicate possible mitochondrial dysfunctions that may trigger compensatory responses in the nucleus. Mitochondria to nucleus retrograde signaling must be triggered by a mitochondrial signal that in turn is relayed to one or more molecules that finally reach the nucleus. The best-known mitochondrial signals are ROS, membrane potential, calcium, ATP and also Mitochondrial-derived peptides recently discovered. Mitochondria-induced changes in calcium fluxes are notably able to activate protein kinase C (PKC), CamKIV, JNK and MAPK, which in turn can activate transcription factors such as ATF2, CEBP/δ, CREB, calcineurin, NFAT and NF-κB. ROS can also activate NF-κB and induce the expression of detoxification enzyme. They can also be responsible for modifying DNA methylation, particularly in cancers. In addition, to the extent that the morphology of mitochondria is highly variable from one tissue to another, this mitochondrial-nucleus cross talk depends on tissues and possible pathologies. The complicating factor is that changes in mitochondrial activity are the cause of disturbances that affect at the same time: membrane potential, ROS, calcium and ATP. In this retrograde integrative signal (black box), it is therefore very difficult to define who is responsible for what. It is therefore almost impossible to predict with certainty the molecular mechanisms that cause changes in the expression of nuclear genes in response to a disruption in mitochondrial activity. Moreover, when looking at a KO animal, one should not lose sight of the fact that the phenotype at a given moment (2, 7, 14 and 3 months) corresponds to an adaptation of the animal to the genetic modification. It is observed in particular for adaptation to oxidative stress.
Concerning p43, it has already been shown that its expression perturbs ROS (Grandemange et al, 2006; Casas et al, 2009), membrane potential (Rochard et al, 2000; Seyer et al, 2006) and calcium fluxes (Saelim et al, 2004). To try to identify the downstream target genes of p43 we performed RNAseq experiments (not yet published) from skeletal muscle of WT mice, p43-/- or overexpressing p43. In muscle, nearly 300 genes out of 12500 have an expression modified more than 2 times in animals p43-/- compared to WT. The mains molecular and cellular functions affected by the deletion of p43 are: Cell-To-Cell Signaling and Interaction, Carbohydrate Metabolism, Small Molecule Biochemistry, Cellular Movement, and Molecular Transport. These data reveal that through the regulation of mitochondrial activity, p43 controls the transcription of numerous previously unidentified genes in skeletal muscle. A similar study is planned using pancreatic isolated islets.
What do you want to do ? New mailCopy